# Employing Data Mining Algorithms and Mathematical Empirical Models for Predicting Wind Drift and Evaporation Losses of a Sprinkler Irrigation Method

**Naji Mordi Naji Al-Dosary** [1,*], **Samy A. Maray** [2], **Saad A. Al-Hamed** [1] and **Abdulwahed M. Aboukarima** [1]

1   Department of Agricultural Engineering, College of Food and Agriculture Sciences, King Saud University, P.O. Box 2460, Riyadh 11451, Saudi Arabia

2   King Saud University, P.O. Box 2460, Riyadh 11451, Saudi Arabia

*   Correspondence: nalsawiyan@ksu.edu.sa

**Abstract:** The advantage of a sprinkler irrigation method is that it saves up to 50% of water consumption during the application of water, as compared to any other surface irrigation system. To assess the behavior of a sprinkler irrigation method, wind drift and evaporation losses (WDEL) are often employed as important parameters. The predictive capacities of four previous mathematical empirical models and two data mining algorithms, namely, reduced-error pruning tree (REPTree) and artificial neural network (ANN) models, were employed to evaluate the impact of the operating parameters of a sprinkler irrigation method on WDEL. The inputs to the REPTree and ANN models were the working pressure, vapor pressure deficit, air temperature, wind speed, nozzle diameter, and air relative humidity. In the experimental field, for data collection, a solid set of sprinklers and collectors positioned per ASAE standards was employed. Promising results showed remarkable performance for one of the mathematical empirical models tested, with a confidence index value of 0.829. Meanwhile, the REPTree and ANN models presented smaller errors for testing data set and are qualified for use given their confidence index values of 0.956 and 0.964, respectively. The REPTree and ANN algorithms were classified as optimal models, indicating that the use of mathematical experimental models alone is inadequate in operational situations involving the nozzle diameter, working pressure, and other variables.

**Keywords:** sprinkler irrigation; wind drift; evaporation losses; reduced-error pruning; multilayer perceptron; Weka software

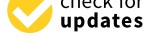



## 1. Introduction

Irrigation water scarcity is a significant issue that has an impact on agricultural production in arid and semi-arid countries [1]. However, fresh water is mostly used for irrigation [2,3]; additionally, irrigated agriculture today accounts for 70–80% of water consumption worldwide [4–7]. In order to reduce the wastage of precious water resources, research on water conservation should be continued using contemporary irrigation technologies, such as sprinkler and trickle irrigation [8], or using correctly designed and maintained irrigation systems [9,10]. Additionally, the development of water conservation techniques requires a thorough understanding of the variables influencing the operation of a sprinkler irrigation system [11].

When compared to surface irrigation methods, sprinkler irrigation methods can cut irrigation water consumption by up to 50%, while also offering the benefits of excellent quality, affordability, and simplicity of installation [12]. A sprinkler irrigation method is not always preferable to drip irrigation or other types of surface irrigation methods. Because the water is transported through pipes, modern pressurized irrigation techniques, such as sprinkler and trickle watering, help reduce water waste [13]. Additionally, the sprinkler irrigation method is a recommended technique in arid and semi-arid environments, owing

to the lack of irrigation water [14]. A portion of the irrigation water delivered by the sprinkler nozzles during sprinkler irrigation operation evaporates before reaching the soil, and this is lost water. Interception losses and wind drift and evaporation losses (WDEL) are two categories that apply to these losses [15]. The performance behavior of a sprinkler irrigation method can be assessed using the parameter WDEL [16]. For the purpose of creating irrigation water management strategies, it is crucial to have a thorough grasp of the variables impacting WDEL in a sprinkler irrigation method. The amount of water that could be lost in semi-arid regions due to wind speed and evaporation would be substantial. WDEL vary from 1.4 to 12.8% of applied water under experimental conditions [17].

The first step in analyzing the WDEL in a sprinkler irrigation method is to undertake an experimental investigation that adheres to a thorough theoretical approach (modeling) [14]. Experimental values of WDEL have ranged from 2 to 50%, as reported in the literature [18]. The climate and hydraulic properties present at the experimental locations are to blame for this variability. As a matter of fact, it has been noted that WDEL are influenced by the wind speed, air relative humidity, pressure head at the nozzle, riser height, air temperature, and drop diameter [11]. Yazar [19] demonstrated that WDEL may comprise a significant portion of the supplied irrigation water in arid and semi-arid regions.

Accurate models for forecasting future water needs are required in order to enhance water management in irrigated areas [5]. Operating conditions, such as working pressure and nozzle diameter, and meteorological factors, such as air temperature, wind speed, vapor pressure deficit, and air relative humidity, have been combined to modify the existing empirical models for WDEL calculation [11,19–23].

Due to the complexity of the application of mathematical modeling of WDEL during irrigation—which is influenced by numerous elements that have been required—a number of simplifications of WDEL mathematical modeling have been presented [24]. The application of empirical models should, according to Saraiva et al. [23], be restricted to operational conditions, such as nozzle diameter and working pressure. Nonlinear correlations between WDEL and impact parameters are used in the majority of empirical models. With the assumption that WDEL are a function of water discharge, nozzle diameter, working pressure, air relative humidity, and wind speed, however, Al-Jumaily and Abdul-Kader [25] used a dimensional analysis technique to develop a prediction equation for WDEL. The correlation coefficient was 0.93 between the predicted and actual WDEL, which appears to be acceptable.

Data mining algorithms have been increasingly used in recent years for modeling in the field of hydrology, according to Pulido-Calvo et al. [26], for the development of predictive models to predict different hydrological parameters, such as pan evapotranspiration, flood forecasting, rainfall forecasting, and weather forecasting; however, these algorithms have limited applications for WDEL prediction. By examining intriguing and practical information, data mining extracts fresh and practically relevant information from enormous datasets. These algorithms are strong, adaptable, and promising for future investigations of mutable or susceptible phenomena [27]. The approach of default equations is not used in data mining algorithms, which is the primary distinction between data mining algorithms and statistics. Statistics experts look for equations that fit the defaults for the majority of the statistical procedures used [28].

There are numerous uses for data mining methods in the agricultural sector. The capacity of 10 Weka data mining models to forecast monthly potential evapotranspiration for upcoming months was tested by Mirhashemi and Tabatabayi [29]. REPTree, Additive Regression, Bagging, M5P, Kstar, Linear Regression, Zero, and M5Rules are examples of data mining algorithms that can be employed for modeling purposes. The dew point, relative humidity, average wind speed, average temperature, daylight hours, and saturation vapor pressure deficit were used as the inputs. The statistical indices indicate that Tree Bagging models are more accurate in determining the monthly average temperature. Teixeira et al. [30] used Weka software to predict the amount of organic matter and clay content in the soil by applying the different data mining algorithms available in it. With

better determination coefficients and fewer errors, the results suggested that the Lazy Kstar method has stronger potential for data mining. Terzi [31] used data mining models to estimate rainfall; Terzi [32] employed a data mining model to predict daily pan evaporation, while Keskin et al. [33] created, using data mining algorithms, a model to predict daily pan evaporation and demonstrated that the REPTree model had greater agreement with actual daily pan evaporation than other models. Artificial neural networks (ANNs) have found widespread use in modeling and simulation due to their high performance and capacity to identify nonlinear complex correlations between the input and output variables of a system [34]. ANNs were employed to predict the reference evapotranspiration [35,36].

The quantification of WDEL is of great significance, both economically and environmentally. It may be possible to minimize WDEL in sprinkler irrigation systems by quantification; however, the estimation of WDEL is very complex, owing to the difficulties encountered in the techniques used to measure such losses [22]. Therefore, this investigation was established to evaluate the impact of different parameters for a sprinkler irrigation method on WDEL for a single sprinkler type (a model by Riegos Costa 130 H—RC130-BY). Moreover, the other goal was to evaluate the ability of seven previous mathematical empirical models and two data mining algorithms to estimate WDEL using experimental data. The data mining algorithms were the reduced-error pruning tree (REPTree) and an ANN of the multilayer perceptron type. The accuracy of the calculated WDEL investigated by means of empirical and data mining algorithms was also compared according to statistical criteria.

## 2. Materials and Methods

### 2.1. Experimental Site

Field tests were carried out at the experimental educational farm of the College of Food and Agriculture Sciences, King Saud University, in Riyadh, Saudi Arabia. The location's latitude and longitude are 24.67° N and 46.69° E, respectively. The experiments were carried out between February and April 2017. The soil texture in the experimental location was identified as a sandy clay loam. A fixed sprinkler irrigation system provided the water. In all, 81 tests were conducted throughout the day at various times to ensure logical and trustworthy results.

### 2.2. Procedures for Sprinkler Tests

The experiments, which used a solid set of sprinklers fixed at a rectangular spacing of 18 × 18 m, were carried out while taking Merriam and Keller's [37] recommendations into consideration. As shown in Figure 1, the catch cans were distributed in accordance with the ASAE Standard [38], and Table 1 displays the distances between collectors (catching cans) for each determination of the throw radius. At varied values of 200, 300, and 400 kPa for sprinkler working pressures and at a riser height of 2 m, one sprinkler, type RC130-BY, was assessed.

The pluviometers were placed along four mutually perpendicular radii from the sprinkler, and they were employed to obtain the irrigation depth (ID) that was released by the spray. There was a distance of 50 cm between the sprinkler and collector, since a total of 120 pluviometers were set along each radius at 1.5 m above ground level. The three inner diameters of the primary nozzle were under test (4, 4.5, and 5 mm). For all of the experimental tests, the sprinkler's plastic auxiliary nozzle had an inner diameter of 2.5 mm.

The experimental tests were conducted under windy conditions because a dead calm was not possible, and the experimental plot was bordered by windbreaks to lessen the impact of the wind. The experimental tests took 2 h on average to complete. To assess the WDEL, 81 experiments consistent with various arrangements of the three working pressures and three primary nozzle diameters were carried out.

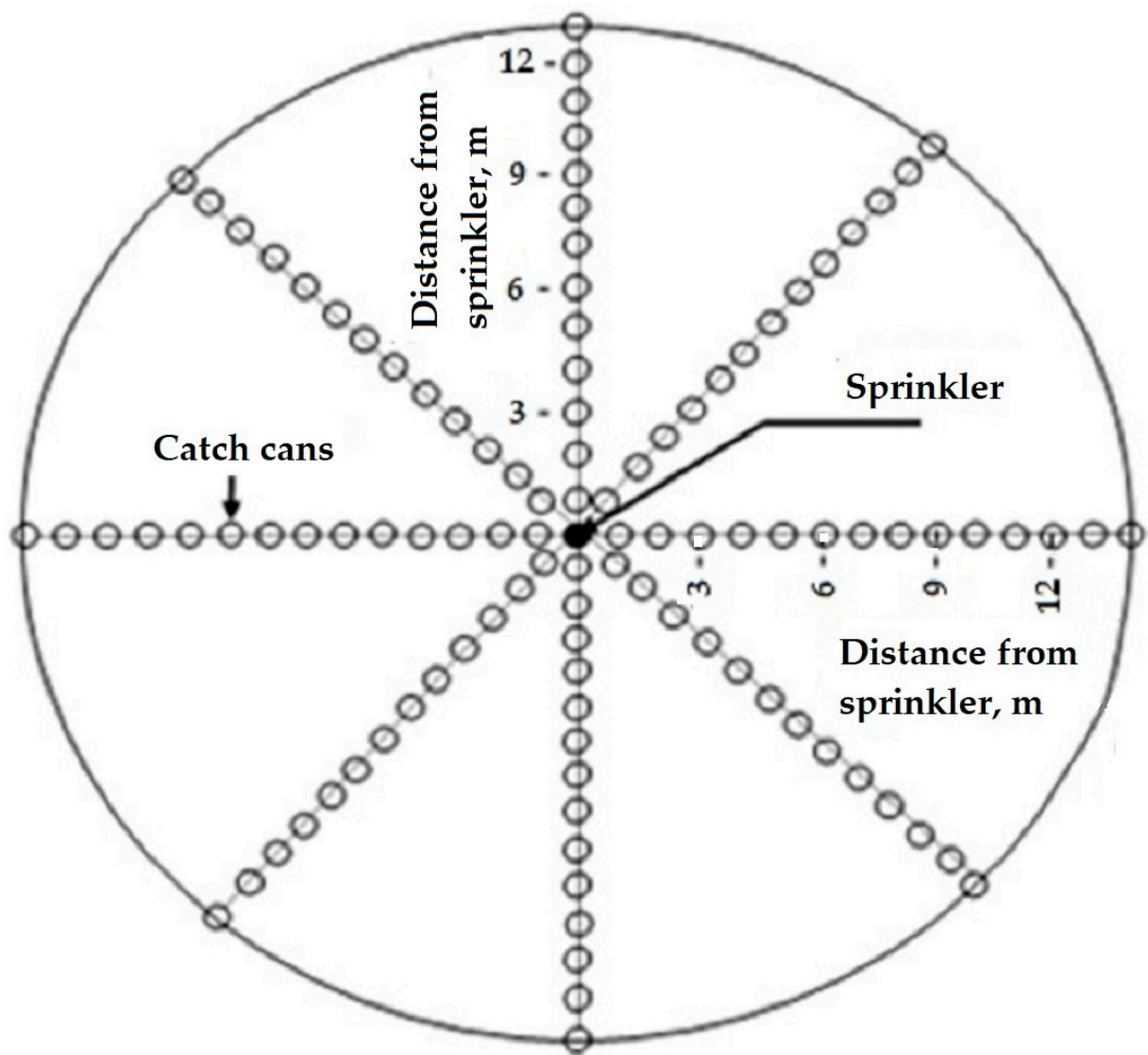

**Figure 1.** Schematic diagram representing the distribution of catch cans.

**Table 1.** Spacing of collectors according to the ASAE Standard [38].

| Sprinkler Throw Distance (m) | Maximum Collector Distance between Centers (m) |
|---|---|
| 0.3–3 | 0.30 |
| 3–6 | 0.60 |
| 6–12 | 0.75 |
| >12 | 1.50 |

A manometer was mounted in the experiment's head control and manual valves were used to manage the working pressure. A meteorological station situated in a plot close to the testing location automatically kept track of the air temperature, air relative humidity, and wind speed throughout the experiments. Averaged records of the climatic variables were used. Using Equation (1), the amount of irrigation water discharged (Q, L/s) was calculated:

$$Q = C_d \times A \times (2g \times P)^q \tag{1}$$

where $C_d$ is the coefficient of discharge ($C_d = 0.98$) according to findings from experiments performed by Playán et al. [39] and Ouazaa et al. [40]. By monitoring the flow rate in

the field, Playán et al. [39] obtained the coefficient of discharge of the RC130-BY sprinkler for a range of working pressures. The nozzle orifice's area is A, the acceleration due to gravity is g (measured in m/s$^2$), the working pressure is P (measured in kPa), and q is a constant [41–43]. For this study, q was taken to be equal to 0.50. In each set test, the irrigation water depth (ID) released by the sprinkler was estimated using Equation (2):

$$\text{ID} = \frac{Q \times t}{S} \tag{2}$$

where Q is the amount of irrigation water discharged (L/s), computed using Equation (1); t is the test time (seconds); and S is the sprinkler spacing, which was determined to be 18 × 18 m in the experimental tests (i.e., 324 m$^2$). The proportion of the irrigation water depth (ID) released by the sprinkler that was not collected in the pluviometers was employed to obtain the percentage of WDEL during each test [18,44,45]. The WDEL were calculated using Equation (3):

$$\text{WDEL} = \frac{(\text{ID} - \text{ID}_{CC}) \times 100}{\text{ID}} \tag{3}$$

where ID is the average water depth that the sprinkler emits, while $\text{ID}_{CC}$ is the average water depth measured by the pluviometers.

### 2.3. WDEL Mathematical Empirical Models

Murray [46] defined the vapor pressure deficit, Δe (kPa), as shown in Equation (4):

$$\Delta e = (e_s - e_a) = 0.611 \times \exp\left(\frac{(17.27 \times T)}{(237.3 + T)}\right) \times \left(1 - \frac{RH}{100}\right) \tag{4}$$

where $e_a$ and $e_s$ are the actual vapor pressure of the air and the saturation vapor pressure, respectively, kPa; T is the air dry-bulb temperature, °C; and RH is the relative humidity of the air, %. The same meteorological and operating conditions were simulated for WDEL using the mathematical empirical models of Yazar [19], Trimmer [20], Tarjuelo et al. [11], and Playán et al. [18]. Table 2 lists the investigated empirical equations, where WDEL are expressed as a percentage (%), D is the primary nozzle diameter expressed in millimeters, Δe is the vapor pressure deficit expressed in kPa, P is the working pressure expressed in kPa, and W is the wind speed expressed in meters per second.

**Table 2.** The investigated previous mathematical empirical models to estimate WDEL.

| Model | Empirical Equation |
|---|---|
| Trimmer [20] | $\text{WDEL} = \left(1.98 \times D + 0.22\Delta e^{0.63} + 3.6 \times 10^{-4} \times P^{1.16} + 0.14 \times W^{0.7}\right)$ |
| Yazar [19] | $\text{WDEL} = (0.003 \times \exp(0.2 \times W) \times \left(10 \times \Delta e \times 10^{0.59} \times T^{0.23} \times P^{0.76}\right) + 0.2$ |
| Tarjuelo et al. [11] | $\text{WDEL} = (0.007 \times P + 7.38 \times \Delta e^{0.5} + 0.844 \times W)$ |
| Playán et al. [18] | $\text{WDEL} = \left(20.3 + 0.214 \times W^2 - 2.29 \times 10^{-3} \times RH^2\right)$ |

### 2.4. Details of the Data Mining Algorithms

The data mining algorithms used in this study were implemented in Weka [47]. Weka is a Java program that is free to download from the website. The algorithms used were ANN and REPTree models. The algorithms were trained using 81 data points by using a percentage split of the collected experimental data (80% for training and 20% for testing). The study inputs were the working pressure (P), nozzle diameter (D), air temperature (T), air relative humidity (RH)H), vapor pressure deficit (Δe), and wind speed (W).

### 2.5. Multilayer Perceptron

According to Bishop [48], a Multilayer Perceptron (MLP) is an artificial neural network that has been trained via backpropagation. A directed link from lower neurons to a neuron in a higher layer is formed by feed-forward connections between the layers of computing units that make up an MLP. An MLP's fundamental building blocks are an input layer, one or more hidden layers, and an output layer. The output of neurons in the hidden layer is what gives them their name; it is only used within the network and is not visible to outside observers. Figure 2 displays an ANN model with a single hidden layer that can forecast the WDEL. Units use an output from one unit in the layer below as an input. There is a weight attached to each connection between units in successive layers.

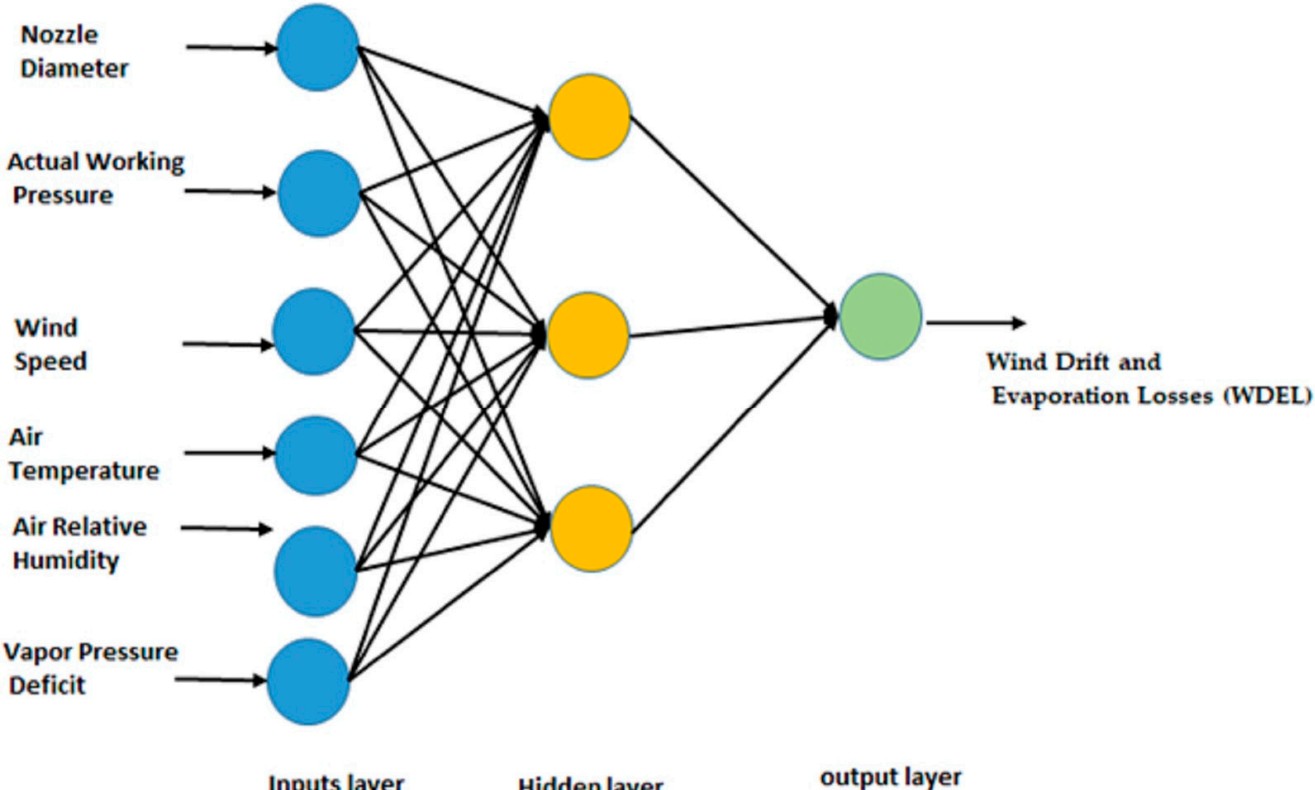

**Figure 2.** An ANN model with one hidden layer to predict WDEL. Each connection is linked with a weight. Sigmoid units are employed as hidden and output units (number of epochs were 500, error per epoch was 0.0072431, learning rate was 0.3, and momentum was 0.2).

The hidden and output units are based on sigmoid units. Input is combined linearly by a sigmoid unit, which then applies the sigmoid function to the output. In Equation (5), the sigmoid transfer function for net input x is condensed.

$$\text{Sigmoid } (x) = \frac{1}{(1 + e^{-x})} \tag{5}$$

The backpropagation algorithm is used by an ANN model to learn its weights [49]. For the purpose of learning, the backpropagation method uses a set of training cases. The weights are set to small random numbers for the specific feed-forward network. Every training example is sent into the network, and each unit's output is computed. To determine the error, the network compares the target output to the output it computed, and the error value is then sent back into the network. Backpropagation employs gradient descent to minimize the squared error between the goal output and the computed output in order to modify the weights. Each unit in the network uses its error value to modify the weights

of its connections, starting with the output unit and working down to the hidden units. Equation (6) is used to update the weights:

$$w_{ji} = w_{ji} + \alpha \times \delta_j \times x_{ji} \tag{6}$$

where $x_{ij}$ is the input from unit i to j, $w_{ji}$ is the weight from unit i to j, $\alpha$ is the learning rate, and $\delta_j$ is the error discovered at unit j. The weights are adjusted using training examples, and this procedure is repeated a certain number of cycles or until the inaccuracy is negligible or cannot be lowered. The weight update at the $n^{th}$ iteration of the backpropagation is made partially dependent on the amount of weight changed in the $(n-1)^{th}$ iteration in order to increase the performance of the backpropagation process. A constant termed the momentum term ($\beta$) controls how much the $(n-1)^{th}$ iteration contributes; however, it is increased to produce a quicker convergence. Equation (7) provides the new rule applied for the weight update at the $n^{th}$ iteration.

$$\Delta w_{ji}(n) = \alpha \times \delta_j \times x_{ji} + \beta \times \Delta w_{ji}(n-1) \tag{7}$$

### 2.6. REPTree

The reduced-error pruning tree (REPTree) is a quick decision tree learning algorithm. It utilizes information gain and variance to construct a decision/regression tree and reduced-error pruning to prune it (with back fitting). For numeric attributes, it only sorts values once. In order to handle missing values, the associated instances are divided into pieces (Weka Software). We refer the reader to Witten and Frank [47] for more information on the REPTree. Weka provides different options to apply the REPTree for modeling and, in this study, default data were selected.

### 2.7. Prediction Performance of Fitted Models

The suitability of the WDEL results, which were also produced using several simulation models for sprinkler irrigation systems in accordance with Conceição and Coelho's guidelines [50], was assessed. According to Willmott [51], the confidence index (c) of Camargo and Sentelhas [52] is created by multiplying the correlation coefficient (r) by the index of agreement (d). The effectiveness of the confidence index (c) was assessed using a scale suggested by Camargo and Sentelhas [52]. Using Equation (8), the confidence index (c) was calculated.

$$c = r \times d \tag{8}$$

The performance confidence index was evaluated according to the classification of Camargo and Sentelhas [52]: optimal ($c > 0.85$); very good $0.76 \leq c \leq 0.85$; good ($0.66 \leq c \leq 0.75$); average ($0.61 \leq c \leq 0.65$); tolerable ($0.51 \leq c \leq 0.60$); bad ($0.41 \leq c \leq 0.50$); and terrible ($c \leq 0.40$).

Using Equation (9), the index of agreement (d) was calculated and using Equation (10), the correlation coefficient (r) was calculated.

$$d = 1 - \left[ \frac{\sum(\hat{Y}_i - Y_i)^2}{\sum(|\hat{Y}_i - \overline{Y}| + |Y_i - \overline{Y}|)^2} \right] \quad 0 \leq d \geq 1 \tag{9}$$

$$r = \frac{\sum_{i=1}^{Nt}(Y_i - \overline{Y}) \times \left(\hat{Y}_i - \overline{\hat{Y}}\right)}{\sqrt{\sum_{i=1}^{Nt}(Y_i - \overline{Y})^2} \times \sqrt{\sum_{i=1}^{Nt}\left(\hat{Y}_i - \overline{\hat{Y}}\right)^2}} \tag{10}$$

Here, $\hat{Y}$ is the predicted or estimated WDEL by the investigated models, $Y_i$ is the value of WDEL observed in field experiments, $\overline{Y}$ and $\overline{\hat{Y}}$ are the means of the observed and predicted WDEL values, and Nt is the number of data points in the testing data set.

The accuracy of the chosen predictive models was evaluated using statistical criteria, such as the mean absolute error (MAE) and root mean square error (RMSE) (Equa-

tions (11) and (12)). The correlation coefficient, in particular, measures the degree of statistical agreement between the target variable's estimated and actual values. This coefficient has a value of 0 when there is no connection and ranges from +1 (the ideal case of perfect direct correlation) to −1 (perfect inverse correlation). For reasonable prediction algorithms, negative values should not happen. The RMSE runs from 0 (the ideal scenario) to ∞ (infinity) and is measured in the same unit as the dependent variable. The average of errors without their sign is equivalent to the mean absolute error [47].

$$\text{RMSE} = \sqrt{\left(\frac{1}{\text{Nt}}\right) \times \sum_{i=1}^{\text{Nt}} (\hat{Y}_i - Y)^2} \tag{11}$$

$$\text{MAE} = \left(\frac{1}{\text{Nt}}\right) \times \sum_{i=1}^{\text{Nt}} |\hat{Y}_i - Y_i| \tag{12}$$

## 3. Results and Discussion

### 3.1. Wind Drift and Evaporation Losses (WDEL)

To analyze the WDEL from the RC130-BY sprinkler irrigation system, various tests were run with a single nozzle in the field. Table 3 presents the average data for operational and meteorological variables, such as the nozzle diameter, temperature, wind speed, and relative humidity, and their influence on WDEL.

**Table 3.** WDEL averaged across a range of working pressures, nozzle diameters, and environmental factors.

| Nozzle Diameter | Actual Working Pressure | Wind Speed | Air Temperature | Air Relative Humidity | Vapor Pressure Deficit | WDEL |
|---|---|---|---|---|---|---|
| (mm) | (kPa) | (m/s) | (°C) | (%) | (kPa) | (%) |
| 4 | 188.1 | 0.82 | 14.93 | 57.11 | 0.73 | 11.60 |
| 4 | 286.6 | 1.07 | 19.12 | 47.89 | 1.16 | 14.85 |
| 4 | 379.4 | 1.27 | 21.59 | 40.56 | 1.54 | 18.49 |
| Overall mean | | 1.05 | 18.55 | 48.52 | 1.14 | 14.98 |
| 4.5 | 191.3 | 0.92 | 15.17 | 59.44 | 0.70 | 11.17 |
| 4.5 | 287.5 | 1.87 | 17.97 | 51.33 | 1.02 | 14.17 |
| 4.5 | 384.5 | 2.85 | 24.89 | 38.44 | 1.95 | 17.94 |
| Overall mean | | 1.88 | 19.34 | 49.74 | 1.22 | 14.43 |
| 5 | 190.4 | 0.87 | 10.83 | 59.22 | 0.53 | 10.61 |
| 5 | 287.3 | 1.82 | 14.73 | 49.11 | 0.86 | 13.68 |
| 5 | 379.8 | 2.58 | 16.71 | 35.67 | 1.23 | 16.25 |
| Overall mean | | 1.76 | 14.09 | 48.00 | 0.87 | 13.52 |

The WDEL increased from 11.6% to 18.49% for a nozzle diameter of 4 mm as working pressure, air temperature, and wind speed rose but relative air humidity fell. For additional nozzle sizes of 4.5 and 5 mm, the same trends and conditional changes were observed. For a nozzle diameter of 4.5 mm, the WDEL increased from 11.17% to 17.94%, and for a nozzle diameter of 5 mm, they increased from 10.61% to 16.25% (Table 3). According to several research publications, operating and meteorological conditions, particularly wind speed, have an impact on WDEL values [10,25,53,54]. The results from Bishaw and Olumana [8] indicated that the threshold value for WDEL in a sprinkler irrigation system was <20%. Additionally, with a solid-set system during day and night irrigation, Playán et al. [18] recorded WDEL of 15.4% and 8.5%, respectively. Fortunately, at all nozzle sizes and working pressures, the results of our study fell inside the range of threshold values (<20%, as reported by [8]). For data associated with a nozzle diameter of 4 mm, the overall mean values of wind speed, air temperature, relative humidity, and vapor pressure deficit were 1.05 m/s, 18.55 °C, 48.52%, and 1.14 kPa, respectively (Table 3). For data associated

with a nozzle diameter of 4.5 mm, the overall mean values of wind speed, air temperature, air relative humidity, and vapor pressure deficit were 1.88 m/s, 19.34 °C, 49.74%, and 1.22 kPa, respectively (Table 3). For data associated with a 5 mm nozzle diameter, the overall mean values of wind speed, air temperature, air relative humidity, and vapor pressure deficit were 1.76 m/s, 14.09 °C, 48.00%, and 0.87 kPa, respectively (Table 3). The numbers in Table 3 also show that, when air temperatures are high and air relative humidity is low, the WDEL will be at their highest. However, we should be aware that, when the air relative humidity level is high while air temperatures are low, the WDEL will also be low. Additionally, regardless of the air temperature, the WDEL will be low when the relative humidity is high. As wind speed and droplet size increase, WDEL rise; this is consistent with the findings of Alnaizy and Simonet [24].

### 3.2. Prediction of WDEL—Data Mining Models

Weka software version 3.6.13 was employed to create an ANN model [47], and training data (full field data) were applied and distributed by Weka into 80% for training and 20% for testing. The Weka graphical user interface (GUI) was used to gather the following data for this study on an irrigation system: the package functions of the weka.classifiers.MultilayerPerceptron -L 0.3 -M 0.2 -N 500 -V 0 -S 0 -E 20 -H a (which can handle sequences), where -L indicates the learning rate, -M indicates the momentum, -N indicates the number of training cycles, -V indicates the validation set size, -S indicates the seed value used by the random number generator (random values are used for initializing weights), -E indicates the validation threshold, and -H indicates the number of hidden layers, with its value "a" representing (num_attributes+num_classes)/2 layers. The value of Num_classe was 0, and the time taken to create the model was 0.29 s. As seen in Figure 2, the default setting of three nodes for the hidden layer was used. In Figure 2, it can be seen that, when the output was compared to the measured WEDL values, the network with three neurons in the hidden layer showed the lowest error of 0.0072431. The values of the error parameters for the ANN model with a structure of three nodes in the hidden layer are presented in Table 4.

**Table 4.** Error statistics regarding WDEL estimation by the REPTree and ANN models using testing data set.

| The Tested Model | RMSE (%) | MAE (%) |
|:---:|:---:|:---:|
| ANN | 0.771 | 0.600 |
| REPTree | 0.679 | 0.544 |

Table 4 compares the error statistics for the ANN and REPTree models for WDEL estimation using the testing data set. The ANN model performed somewhat worse than the REPTree model in terms of accuracy, with RMSE and MAE values of 0.771% and 0.600%, respectively. Figure 3 presents a scatter diagram comparing the ANN-calculated values to the actual values for the testing data set, showing that the ANN model offered a respectable determination coefficient ($R^2$). The points in this figure are sparsely distributed around the regression line ($R^2 = 0.967$), indicating that the values derived from the experimental data are either overestimated or underestimated. The distribution of points around the best fit line demonstrates the method's great accuracy in estimating low WDEL values. Weka software version 3.6.13 was also used to create a REPTree model using the same data used to train the ANN model [47]. Figure 4 displays a scatter plot of the estimated WDEL values from the REPTree model against the values of the tested data that were actually observed. A proper agreement is indicated by the $R^2$ score of 0.943. When predicting low WDEL values, the distribution of points around the best fit line illustrates how inaccurate this method is; nevertheless, as WDEL increase, the amount of error diminishes.

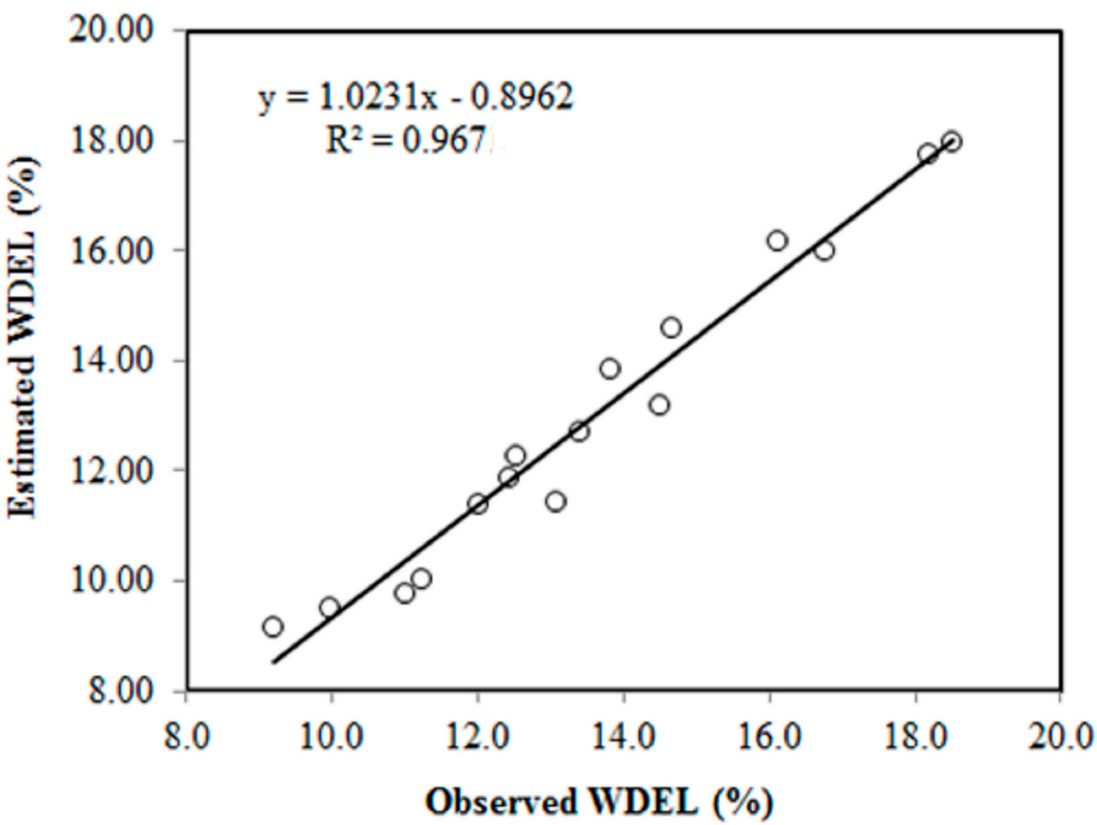

**Figure 3.** Scatter plot of estimated WDEL values using ANN model against observed values of testing data set.

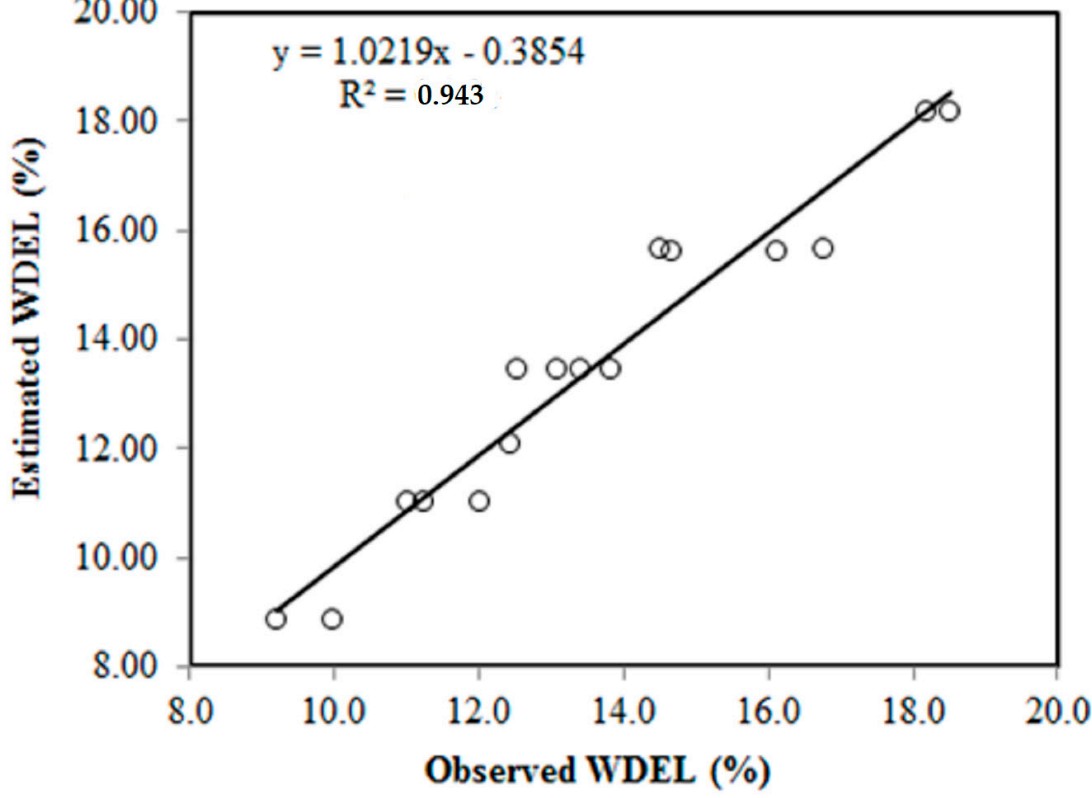

**Figure 4.** Scatter plot of estimated WDEL values using REPTree model against observed values of testing data set.

### 3.3. Mathematical Empirical Models for WDEL Simulations

Table 5 lists the outcomes of the WDEL simulations performed using the exploratory variables of working pressure, nozzle diameter, air temperature, vapor pressure deficit, wind speed, and air relative humidity for the testing data set by the empirical models of Yazar [19], Trimmer [20], Tarjuelo et al. [11], and Playán et al. [18], as well as our ANN and REPTree models. The average WDEL results from Yazar [19], Trimmer [20], and Tarjuelo et al. [11] were 3.22%, 5.99%, and 10.06%, respectively. Additionally, the average WDEL values produced by the Playán et al. [18], REPTree, and ANN models were 14.54%, 13.50%, and 13.0%, respectively. The testing data's equivalent observed WDEL were 13.58%. With the exception of the empirical model [18], which is constrained by the operational conditions, the latter models, therefore, showed lesser errors for testing data set and are better equipped to be utilized in predicting WDEL (nozzle diameter, operating pressure, etc.).

**Table 5.** Observed and predicted WDEL values using different approaches under various climatic and operating conditions (testing data set).

| | | | | | | | Predicted WDEL | | | | | |
|---|---|---|---|---|---|---|---|---|---|---|---|---|
| D | P | W | T | RH | Δe | Observed WDEL | Trimmer [20] | Yazar [19] | Tarjuelo et al. [11] | Playán et al. [18] | REPTree | ANN |
| (mm) | (kPa) | (m/s) | (°C) | (%) | (kPa) | (%) | (%) | (%) | (%) | (%) | (%) | (%) |
| 4.5 | 193 | 1.20 | 15.40 | 56 | 0.77 | 12.44 | 1.99 | 3.75 | 8.84 | 13.43 | 12.104 | 11.91 |
| 4.5 | 288 | 1.98 | 19.11 | 49 | 1.13 | 14.65 | 3.99 | 8.16 | 11.53 | 15.64 | 15.638 | 14.63 |
| 4.5 | 195 | 0.47 | 14.30 | 64 | 0.59 | 9.20 | 1.35 | 2.49 | 7.42 | 10.97 | 8.885 | 9.18 |
| 5.0 | 272 | 1.43 | 14.63 | 51 | 0.82 | 12.51 | 2.43 | 5.23 | 9.78 | 14.78 | 13.473 | 12.30 |
| 4.5 | 194 | 1.18 | 14.50 | 58 | 0.69 | 12.00 | 1.90 | 3.49 | 8.50 | 12.89 | 11.037 | 11.42 |
| 4.0 | 186 | 0.80 | 16.90 | 56 | 0.85 | 13.08 | 2.17 | 3.41 | 8.77 | 13.26 | 13.473 | 11.45 |
| 5.0 | 388 | 2.57 | 16.81 | 36 | 1.23 | 16.76 | 5.57 | 12.10 | 13.06 | 18.75 | 15.683 | 16.03 |
| 5.0 | 189 | 0.93 | 10.51 | 60 | 0.51 | 11.01 | 1.26 | 2.47 | 7.37 | 12.24 | 11.037 | 9.80 |
| 4.5 | 299 | 1.89 | 17.50 | 52 | 0.96 | 13.81 | 3.77 | 7.35 | 10.92 | 14.87 | 13.473 | 13.86 |
| 5.0 | 195 | 0.95 | 11.24 | 59 | 0.55 | 11.23 | 1.33 | 2.68 | 7.63 | 12.52 | 11.037 | 10.05 |
| 4.0 | 379 | 1.30 | 22.10 | 41 | 1.57 | 18.50 | 6.18 | 9.91 | 13.00 | 16.81 | 18.204 | 17.98 |
| 4.0 | 376 | 1.28 | 20.50 | 40 | 1.45 | 18.18 | 5.86 | 9.21 | 12.59 | 16.99 | 18.204 | 17.79 |
| 4.0 | 290 | 1.02 | 19.10 | 53 | 1.04 | 13.40 | 3.59 | 5.80 | 10.42 | 14.09 | 13.473 | 12.71 |
| 4.0 | 285 | 0.87 | 19.46 | 49 | 1.15 | 14.50 | 3.53 | 5.84 | 10.65 | 14.96 | 15.683 | 13.20 |
| 4.5 | 190 | 0.52 | 14.70 | 62 | 0.64 | 9.96 | 1.40 | 2.62 | 7.65 | 11.56 | 8.885 | 9.52 |
| 5.0 | 379 | 2.43 | 16.32 | 34 | 1.23 | 16.10 | 5.26 | 11.35 | 12.87 | 18.92 | 15.638 | 16.19 |
| Average | | | | | | 13.58 | 3.22 | 5.99 | 10.06 | 14.54 | 13.50 | 13.00 |
| Minimum | | | | | | 9.20 | 1.26 | 2.47 | 7.37 | 10.97 | 8.89 | 9.18 |
| Maximum | | | | | | 18.50 | 6.18 | 12.10 | 13.06 | 18.92 | 18.20 | 17.98 |
| Standard deviation | | | | | | 2.75 | 1.75 | 3.30 | 2.11 | 2.40 | 2.89 | 2.86 |

For the testing data set (Table 6, c = 0.314), the model suggested by Trimmer [20] to predict the WDEL of the examined sprinkler displayed a "Terrible" performance index, estimating smaller WDEL values than those that actually occurred. When the Trimmer [20] model was used to analyze the data by Beskow et al. [22], a poor performance index was likewise demonstrated in their study. The nozzle diameters and pressure ranges for which the Trimmer model is appropriate can be blamed for the model's limitations. For testing data set (Table 6, c = 0.393), the model suggested by Yazar [19] to forecast the WDEL of the examined sprinkler showed a "Terrible" performance index, estimating smaller WDEL values than those that actually occurred. Table 5 further demonstrates that Yazar [19] underestimated WDEL, which was previously noted in research by Beskow et al. [22]. For testing data set, the Tarjuelo et al. [11] model provided an "Average" performance indicator (Table 6, c = 0.617), estimating slightly lower WDEL values than those that actually occurred.

However, the respective standard deviations of the observed data and data from Tarjuelo et al. [11] were 2.75% and 2.11%. The Playán et al. [18] model estimated a maximum value of 18.92% and provided a "Very good" performance index for the testing data set (Table 6, c = 0.829). The maximum value for the measured WDEL was 18.50%; this is due to the likelihood that the results would have been used for a sprinkler system with a comparable design (nozzle diameter and working pressure range). As fantastic models, REPTree and ANN displayed confidence index values of 0.956 and 0.964, respectively, in Table 6, demonstrating that the use of empirical models is restricted to operational settings (nozzle diameter, working pressure, etc.). Due to the ANN model's greatest correlation coefficient values, the confidence index for ANN model was greater than that for REPTree model.

**Table 6.** Results of the index of agreement (d), correlation coefficient (r), and confidence index (c) tests for the different prediction methods in relation to observed WDEL using the testing data set.

| Prediction Method | Index of Agreement | Correlation Coefficient | Confidence Index | Performance Based on Confidence Index |
|---|---|---|---|---|
| The model described by Trimmer [20] | 0.325 | 0.966 | 0.314 | Terrible |
| The model described by Yazar [19] | 0.437 | 0.898 | 0.393 | Terrible |
| The model described by the model described by Tarjuelo et al. [11] | 0.650 | 0.949 | 0.617 | Average |
| Playán et al. [18] | 0.913 | 0.908 | 0.829 | Very good |
| REPTree model | 0.984 | 0.971 | 0.956 | Optimal |
| ANN model | 0.980 | 0.983 | 0.964 | Optimal |

## 4. Conclusions

When compared to the use of any other surface irrigation method, the sprinkler irrigation method has the advantage of using less water. However, before reaching the soil in the sprinkler irrigation method, some of the water emitted by the nozzles is lost to wind drift and evaporation losses (WDEL). According to reports, WDEL values range from 2% to 50% and are influenced by factors such as riser height, vapor pressure deficit, air relative humidity, wind speed, air temperature, working pressure, and nozzle diameter. In arid and semi-arid environments, WDEL may make up a significant portion of the water delivered. Therefore, using a local sprinkler, this study examined the impact of the operating pressure and nozzle diameter on WDEL. Furthermore, for the sustainable management of irrigation water, a reliable and accurate WDEL forecast model is essential. The novelty of the paper lies in estimating WDEL using the empirical and data mining models, such as the multilayer perceptron neural network and REPTree. The average observed value of WDEL was 13.58%; meanwhile, the predicted average values of WDEL using the investigated empirical models were 3.22%, 5.99%, 10.06%, and 14.54% and they were 13.50% and 13.00% using REPTree and multilayer perceptron neural networks models, respectively. Overall, the multilayer perceptron neural network outperformed the other models on the testing data set. The practical achievement of this research is that choosing a simulation model to predict WDEL is important in traditional sprinkler irrigation systems.

**Author Contributions:** Conceptualization, N.M.N.A.-D., S.A.A.-H. and A.M.A.; methodology S.A.M., S.A.A.-H. and N.M.N.A.-D.; software, A.M.A., S.A.A.-H. and N.M.N.A.-D.; formal analysis, A.M.A., S.A.M. and N.M.N.A.-D.; validation, N.M.N.A.-D., A.M.A. and S.A.M.; visualization, N.M.N.A.-D. and A.M.A.; investigation, N.M.N.A.-D., S.A.A.-H. and A.M.A.; resources, N.M.N.A.-D., S.A.M. and A.M.A.; data curation, S.A.M. and N.M.N.A.-D.; writing—original draft preparation, N.M.N.A.-D., A.M.A., S.A.A.-H. and S.A.M.; supervision, N.M.N.A.-D. and S.A.M.; funding acquisition, S.A.M., N.M.N.A.-D. and A.M.A.; writing—review and editing, all authors. All authors have read and agreed to the published version of the manuscript.

**Funding:** This research was funded by Researchers Supporting Project number (RSPD2023R752), King Saud University, Riyadh, Saudi Arabia.

**Institutional Review Board Statement:** Not applicable.

**Informed Consent Statement:** Not applicable.

**Data Availability Statement:** Data are contained within the article.

**Acknowledgments:** The authors would like to extend their sincere appreciation to the Researchers Supporting Project (RSPD2023R752) King Saud University, Riyadh, Saudi Arabia.

**Conflicts of Interest:** The authors declare no conflict of interest.

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
