# Peer review of "Employing Data Mining Algorithms and Mathematical Empirical Models for Predicting Wind Drift and Evaporation Losses of a Sprinkler Irrigation Method"

_water, doi:10.3390/w15050922_

Round 1
Reviewer 1 Report
Introduction: I suggest to rewrite completely the introduction. In this form it is unreadable, a lot of syntax errors (see follow comments) and sentences not clear. The aim of the paper have to be clarify.
L.33 However??
L. 38-40 Rewrite the sentence.
L.67 metrological???
L.71 “that have been identified”???
L.83 paramters???
L.84 orecasting??
L.85 limeted
L.103 estimate
L.116 paramtres?
L.132 “As a result, a variety of climatic situations were covered” How??? Why??
L.150 “dead calm” is not appropriate for a research article.
L.164 rewrite the two sentences.
Q in the equation 1 is the sprinkler’s discharge, in the equation 2 Q is the amount of irrigation water discharged. The definition have to be the same.
L.168 18m x 18 m
L.188 Two points.
Section 2.5. In my opinion this section is unreadable.
L.249 filed???seen???
L.255 no correlation, not connection.
L.257 Why?
L.271 Went???
L.279 “the results of our study fell inside the range of the threshold values”What??
Figure 4 and Figure 5. The black line is the regression line??? It seems a linear regression without the intercept, but in the relationship that you reported in the figures the interception value is set.
L.376 defects???
Author Response
Dear Editor at the Water Editorial Office,
Based on your message and the reviewers' comments, we are submitting the revised manuscript (a research article) for consideration and publication in the Water Journal and we appreciate all of reviewers' comments that will improve the quality of our manuscript. The original scientific manuscript is entitled “Employing Data Mining Algorithms and Mathematical Empirical Models for Predicting Wind Drift and Evaporation Losses of a Sprinkler Irrigation Method,
By Naji Mordi Naji Al-Dosary, Samy A. Maray, Saad A. Al-Hamed and Abdulwahed M. Aboukarima”. Ref. Manuscript ID: water-2200291
At first and before answering all of reviewers' comments that have improved the quality of our manuscript, thank you for your trust in us and giving us a chance to explain based on the reviewers’ responses. We sympathize with their responses, but we will stay completely objective in our comments. We would like to be aware that the reviewer’s response to give an advantage of publishing a good paper about the same subject and I believe the work has value and contributes to the science of mathematical empirical models and data mining algorithms (DMAs).
In recognition of our well-respected work, and which will serve as a useful resource for future studies worth reading and sharing, we hope that you will appreciate our manuscript, and consider it for publication in the journal. We will be very happy to publish it in your journal, also we thank you in advance for your attention.
(Note, to improve the paper quality and language, some changes (track all changes made to the article file) were appeared in the revised manuscript).
Yours sincerely,
Naji Mordi Naji Al-Dosary
(Corresponding author) and all co-authors

Reviewer 2 Report
Dear authors,
the work tries to compare different literature methodologies for the prediction of sprinkler irrigation wind drift and evaporation losses. The comparison is made on the basis of the results of a specific experimental campaign.
Unfortunately your work was very difficult to be understood due to low quality of english writing and to many spelling and grammar errors (see attachment).
The work can be valuable for publication because the topic is very interesting and actual but it requires deep review before new submission.
The paper is well structured but more attention must be paid to the formal presentation of the methodologies used and of the results.
The conclusions must be improved to highlight better the novelty and the new achievement reached by the work.
Attached you can find a pdf with point by point revisions.

Author Response

(The authors gave the same response as above.)

Round 2
Reviewer 1 Report
I read the revised paper and in my opinion, the paper is publishable in this form.
Author Response
Many thanks to the MDPI reviewers for their valuable and helpful comments that would develop our article in a distinctive style.

Reviewer 2 Report
Dear Authors,
thanks for reviewing your work. The english language and style have been improved.
Anyway the paper needs still some updates. You find a pdf file attached with some revisions.
A general comment: Figure 1, Figure 2 and Figure 3 have poor quality. Moreover, usually research paper does not show directly the input windows of a software this kind of figures are more appropriate for a manual or report style work.
Thank you

Author Response

(The authors gave the same response as above.)
